# Murine breast cancers disorganize the liver transcriptome in a zonated manner

Alexis Vandenbon [1,2,11✉], Rin Mizuno[3,4,11], Riyo Konishi[3,11], Masaya Onishi[5,11], Kyoko Masuda[6], Yuka Kobayashi[6], Hiroshi Kawamoto[6], Ayako Suzuki [5], Chenfeng He[3,7], Yuki Nakamura[3,7], Kosuke Kawaguchi[7], Masakazu Toi [7], Masahito Shimizu[8], Yasuhito Tanaka[9], Yutaka Suzuki [5✉] & Shinpei Kawaoka [3,10✉]

The spatially organized gene expression program within the liver specifies hepatocyte functions according to their relative distances to the bloodstream (i.e., zonation), contributing to liver homeostasis. Despite the knowledge that solid cancers remotely disrupt liver homeostasis, it remains unexplored whether solid cancers affect liver zonation. Here, using spatial transcriptomics, we thoroughly investigate the abundance and zonation of hepatic genes in cancer-bearing mice. We find that breast cancers affect liver zonation in various distinct manners depending on biological pathways. Aspartate metabolism and triglyceride catabolic processes retain relatively intact zonation patterns, but the zonation of xenobiotic catabolic process genes exhibits a strong disruption. The acute phase response is induced in zonated manners. Furthermore, we demonstrate that breast cancers activate innate immune cells in particular neutrophils in distinct zonated manners, rather than in a uniform fashion within the liver. Collectively, breast cancers disorganize hepatic transcriptomes in zonated manners, thereby disrupting zonated functions of the liver.

[1] Laboratory of Tissue Homeostasis, Institute for Life and Medical Sciences, Kyoto University, Kyoto 606-8507, Japan. [2] Institute for Liberal Arts and Sciences, Kyoto University, Kyoto 606-8507, Japan. [3] Inter-Organ Communication Research Team, Institute for Life and Medical Sciences, Kyoto University, Kyoto 606-8507, Japan. [4] Department of Gynecology and Obstetrics, Kyoto University Graduate School of Medicine, Kyoto 606-8507, Japan. [5] Graduate School of Frontier Science, The University of Tokyo, Chiba 277-8562, Japan. [6] Laboratory of Immunology, Institute for Life and Medical Sciences, Kyoto University, Kyoto 606-8507, Japan. [7] Department of Breast Surgery, Kyoto University Graduate School of Medicine, Kyoto 606-8507, Japan. [8] Department of Gastroenterology/Internal Medicine, Gifu University Graduate School of Medicine, Gifu 501-1194, Japan. [9] Department of Gastroenterology and Hepatology, Faculty of Life Sciences, Kumamoto University, Kumamoto 860-8556, Japan. [10] Department of Integrative Bioanalytics, Institute of Development, Aging and Cancer (IDAC), Tohoku University, Sendai 980-8575, Japan. [11] These authors contributed equally: Alexis Vandenbon, Rin Mizuno, Riyo Konishi, Masaya Onishi. ✉email: alexis.vandenbon@gmail.com; ysuzuki@edu.k.u-tokyo.ac.jp; kawaokashinpei@gmail.com

Remote cancers affect the liver in various manners thereby disrupting host homeostasis[1–11]. For example, circadian gene expression rhythm is disrupted in a breast cancer model and a genetically induced lung cancer model[3,4]. This lung cancer model also disrupts hepatic glucose metabolism[9]. Fearon and colleagues showed that interleukin-6 derived from pancreatic ductal adenocarcinoma suppresses hepatic ketogenesis[2]. A zebrafish intestinal tumor reduces bile production in the liver[5]. We recently found that remote solid cancers dampen hepatic nitrogen metabolism[8]. Some of these abnormalities are detectable in human cancer patients[12,13]. These metabolic abnormalities in the liver are often accompanied by systemic phenotypes, such as reduced behavioral activity, suggesting the significance of liver alteration in systemic phenotypes caused by remote cancers[1,8]. Of note, the above-described discoveries were made through analyses against whole liver tissues.

It has been unknown whether remote cancers disrupt spatially organized gene expression (i.e., zonated gene expression) in the liver. The liver has a spatially organized tissue structure known as zonation, formed by repetitive hexagonal units called liver lobules[14,15]. The lobules are associated with portal veins and central veins. Portal veins are at the junction of neighboring lobules, supplying nutrient- and oxygen-rich blood to nearby hepatocytes. Those hepatocytes eventually consume nutrients and oxygen. Consistent with this, hepatocytes nearby portal veins express genes critical for active energy metabolism. The consequently exhausted blood is then drained by central veins. In contrast to portal vein-associated hepatocytes, hepatocytes nearby central veins are enriched for xenobiotic metabolism genes. Thus, hepatocytes nearby portal veins and central veins are transcriptionally distinct[14]. Recent advances in spatial transcriptomics enable us to obtain zonated gene expression profiles from the liver both in physiological and disease conditions[16–25]. However, it remains unclear whether cancer-induced liver abnormalities are associated with the disruption in liver zonation.

Here, we use spatial transcriptomics to understand how remote cancers affect liver zonation. Combining spatial transcriptomics and single-cell RNA sequencing (scRNA-seq), we characterize the livers of breast cancer-bearing animals at the spatial and single-cell resolution, finding that murine breast cancers disrupt liver zonation. Our data unravel cell type-specific consequences in the liver due to breast cancer transplantation. This study will be an important basis to understand how breast cancers affect spatially organized gene expression in the liver to disturb the whole liver homeostasis.

## Results

### Visualizing liver zonation using spatial transcriptome.
To understand how cancers affect liver zonation, we performed spatial transcriptomic analyses against the livers of sham-operated and 4T1 cancer-bearing animals. The 4T1 breast cancer model is a commonly used syngeneic mouse cancer model[26]. Transplanted 4T1 cells are invasive, forming cancerous tissues in vivo. We exploited this 4T1 model because we previously showed that 4T1 breast cancers remotely affect gene expression and metabolism in the liver[4,8].

We first confirmed the previously reported zonated expression patterns of Albumin (Alb) and Cyp2e1 in the livers of sham-operated mice (Fig. 1a and Supplementary Fig. 1a, b). It has been known that hepatocytes nearby portal veins are supplied with nutrient- and oxygen-rich blood, and abundantly express Alb. On the other hand, hepatocytes nearby central veins express detoxifying genes including Cyp2e1. Exploratory analysis of spatially variable genes (SVGs) using singleCellHaystack[27] confirmed two subsets of genes with the known expression

patterns of Alb and Cyp2e1, demonstrating that these two genes are expressed in different zones in the liver (Fig. 1a).

To further characterize these zonation patterns, we divided the Visium spots of each image into three sets of equal size: those with the highest Alb expression ($Alb^{high}$), those with the highest Cyp2e1 expression ($Cyp2e1^{high}$), and the remaining one-third (with intermediate Alb and Cyp2e1 expressions). We then calculated "module scores" (average expression levels; see the "Methods" section) using Seurat for sets of genes associated with 2898 Gene Ontology (GO) terms (Supplementary Data 1), and compared the $Alb^{high}$ and the $Cyp2e1^{high}$ zones of the two sham samples (Fig. 1b and Supplementary Fig. 2; Supplementary Data 2). As shown in Fig. 1b, the $Alb^{high}$ zone had higher expression of genes involved in "ATP synthesis coupled electron transport" and "aspartate family amino acid metabolic process." This observation was further confirmed by the uniform manifold approximation and projection (UMAP) plots where the zonated expression of Alb was highlighted (Fig. 1c–e). In contrast, the $Cyp2e1^{high}$ zone had higher expression of genes related to the "xenobiotic catabolic process" and "bile acid and bile salt transport" (Fig. 1b and f–h). We found that a set of pathways that have been reported to be correlated with oxygen availability[28] showed such zonation in our datasets, including "gluconeogenesis", "urea metabolic process", and "glutamine family amino acid biosynthetic process" (Supplementary Fig. 3). These pathways appeared active where oxygen is rich (i.e., $Alb^{high}$ spots). We also found pathways being active where oxygen is poor (i.e., $Cyp2e1^{high}$ spots). These pathways included the "xenobiotic catabolic process" and "bile acid biosynthetic process." Thus, our data confirm that different liver zones conduct distinct metabolic pathways.

### Breast cancers alter spatial transcriptomics in the liver.
We next compared the spatial transcriptome of the livers of sham-operated and cancer-bearing mice in various manners. We found that both sham and cancer-bearing samples retained the zonated expression patterns of Alb and Cyp2e1 (Fig. 2a and Supplementary Fig. 1). This data suggested that distal cancers did not completely abolish zonated gene expression in the liver. However, when we analyzed the datasets via dimensionality reduction using principal component analysis (PCA) and UMAP, sham and cancer-bearing spots appeared nearly completely separated, even after batch effect correction (Fig. 2b, UMAP_2 axis, and Supplementary Fig. 4). This lack of overlap between sham and cancer-bearing samples indicated strong changes in the cell composition and/or gene expression at each spot. In addition, this plot also exhibited a clear separation of $Alb^{high}$ versus $Cyp2e1^{high}$ spots along the UMAP_1 axis, further confirming the zonated gene expression in the liver in our datasets (Fig. 2b).

To comprehensively investigate whether cancer transplantation affects liver zonation, we visualized the module scores of genes involved in distinct biological pathways according to the zonated expression of Alb and Cyp2e1 (Fig. 2c, d and Supplementary Data 3). In the analysis, we defined $Alb^{high}$ spots to be the top 10% spots with the highest Alb expression in each image (the red spot in Fig. 2c). We then calculated mean module scores in these $Alb^{high}$ spots and their closest (the six spots at a distance of 100 μm center-to-center; the purple spots in Fig. 2c) and secondary (the six spots at a distance of about 173 μm center-to-center; the blue spots in Fig. 2c) neighboring spots. We did the same thing for the top 10% spots with the highest Cyp2e1 expression ($Cyp2e1^{high}$ spots). This method validated the zonated gene expression of various metabolic pathways, for example, the "xenobiotic catabolic process" and the "aspartate family amino acid metabolic process" in the liver (Fig. 2d).

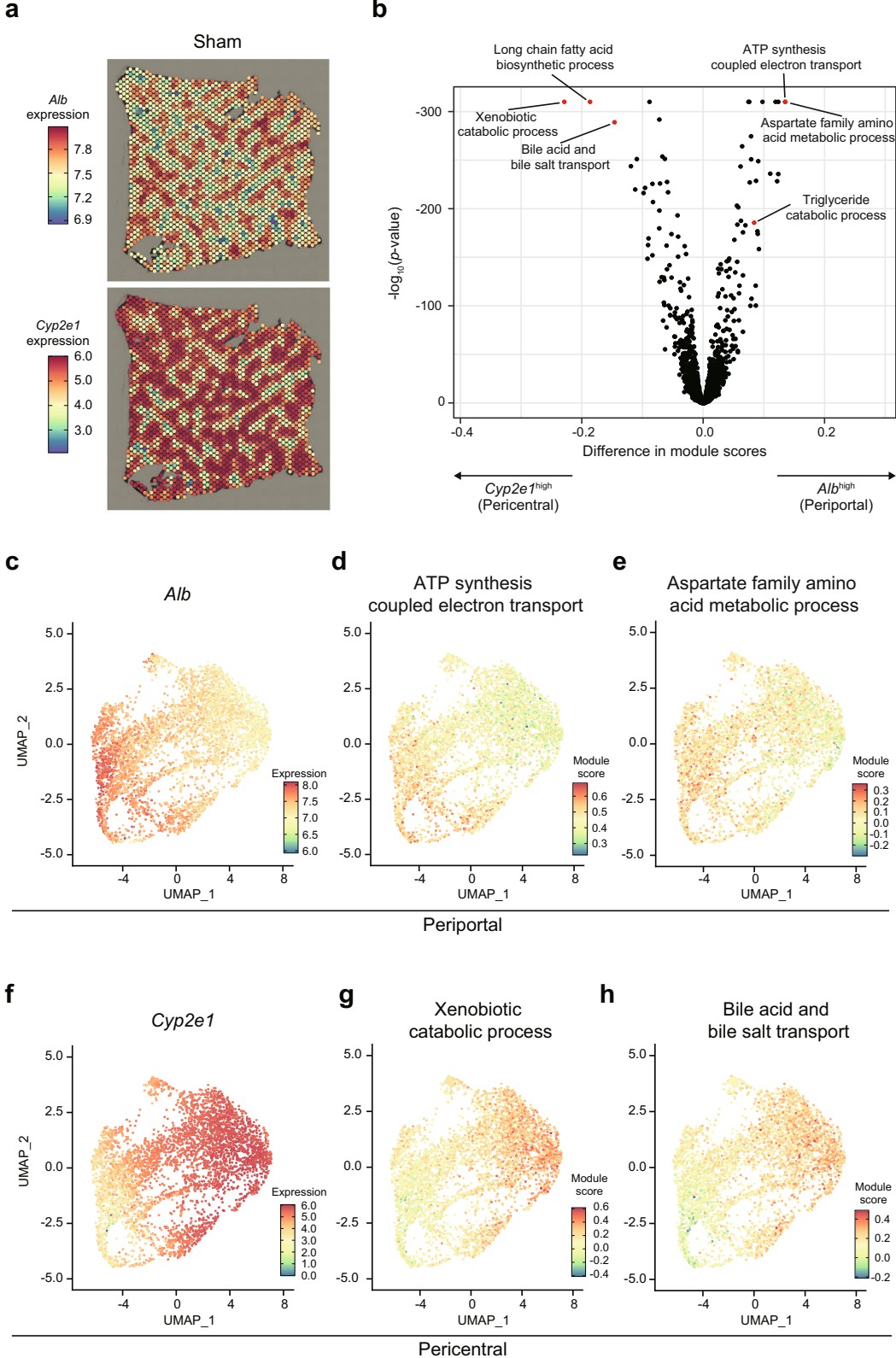

**Fig. 1 Visualization of liver zonation using spatial transcriptomics. a** *Alb* and *Cyp2e1* expressions in one of the sham samples. A second sample is shown in Supplementary Fig. 1. **b** Volcano plot of the module scores of 2898 GO terms. The *x*-axis shows the difference in module scores between *Alb*high and *Cyp2e1*high zones. The *y*-axis shows the *p*-values ($-\log_{10}$) of a Wilcoxon Rank Sum test. Supplementary Fig. 2 shows the same plot with more GO terms indicated. **c**–**h** UMAP plots of the Visium spots of the two sham samples. **c** UMAP_1 is anti-correlated with *Alb* expression. Spots on the left side represent the periportal zone with high *Alb* expression. Genes involved in the ATP synthesis coupled electron transport (**d**) and the aspartate family amino acid metabolism (**e**) have high activity in the periportal zone. **f** UMAP_1 is correlated with *Cyp2e1* expression. Spots on the right side represent the pericentral zone with high *Cyp2e1* expression. Genes involved in the xenobiotic catabolic process (**g**) and the bile acid and bile salt transport (**h**) have high activity in the pericentral zone.

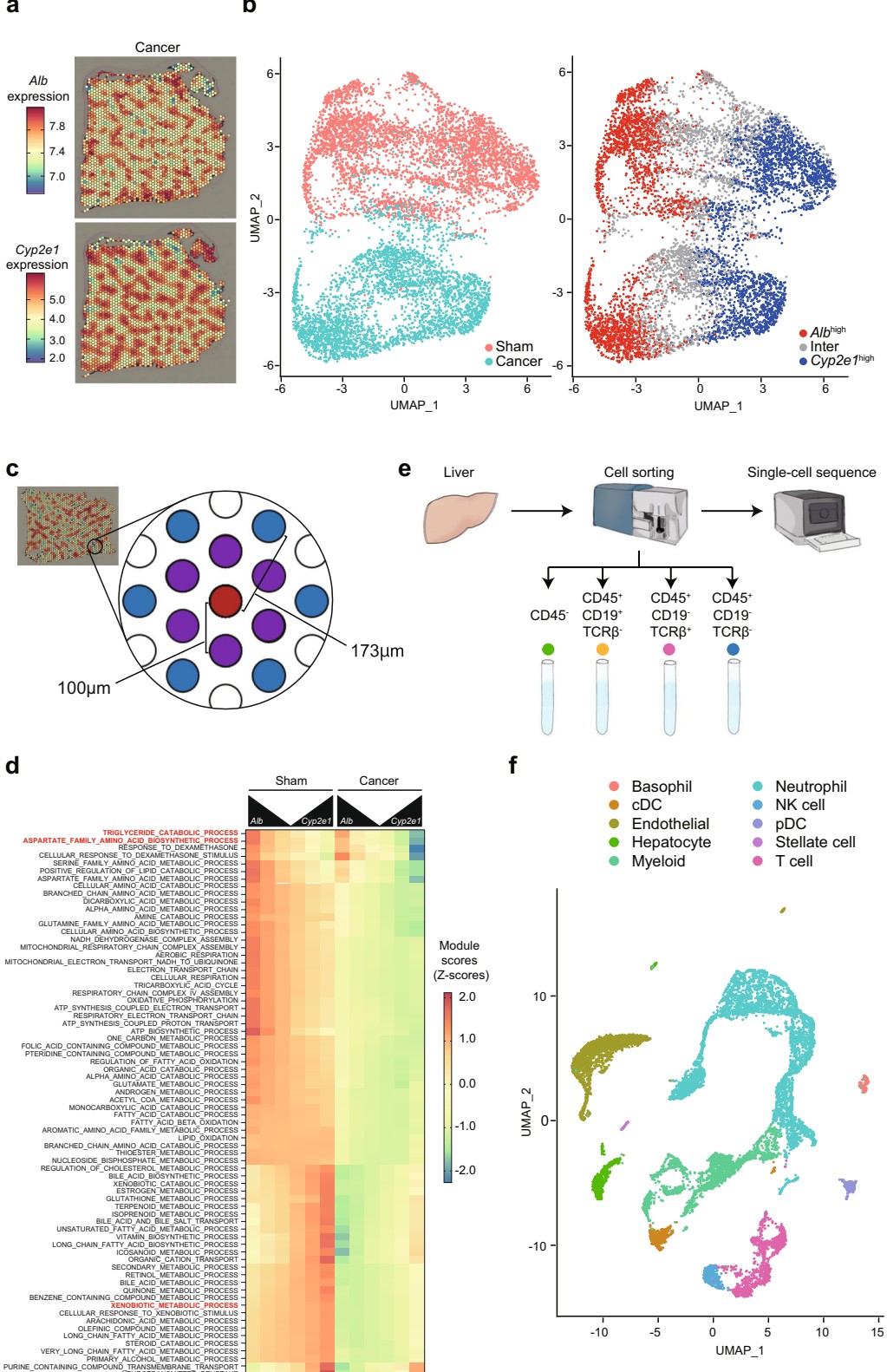

**Fig. 2 Spatial and single-cell characterization of the livers of cancer-bearing mice. a** *Alb* and *Cyp2e1* expression in one of the 4T1 cancer samples. A second sample is shown in Supplementary Fig. 1. **b** UMAP plot of the Visium spots of the two sham and two cancer-bearing samples. Left: spots are colored by condition, revealing a clear distinction between sham and cancer-bearing samples. Right: colors reflect the classification of spots according to *Alb* and *Cyp2e1* expression. **c** Illustration of the definition of closest (purple) and secondary (blue) neighboring spots of a spot of interest (red), and the approximate distances separating them. **d** Heatmap showing the normalized module scores of selected GO terms in the function of their distance to *Alb*high and *Cyp2e1*high spots in sham (left) and cancer-bearing (right) samples. **e** The scheme for single-cell transcriptome. We sorted CD45− cells, CD45+CD19+TCRβ− cells, CD45+CD19−TCRβ+ cells, and CD45+CD19−TCRβ− cells, mixing them at the same proportion among the samples. The obtained mixture of cells is subjected to single-cell transcriptomics. **f** UMAP plot of the scRNA-seq dataset, indicating cell type annotations. Supplementary Fig. 8 shows the same data separated by condition.

We added one more important layer to our analyses. As described earlier, the liver consists of multiple cell types. Given the size of the Visium spots (diameter of 55 µm), it is reasonable to expect that RNA molecules captured at each spot originated from >10 different cells. To address this issue, we performed single-cell transcriptomics analyses (Fig. 2e). We dissociated the whole liver tissues with collagenase and collected $CD45^+CD19^+TCR\beta^-$ B cells, $CD45^+CD19^-TCR\beta^+$ T cells, $CD45^+CD19^-TCR\beta^-$ immune cells that are neither B cells nor T cells, and other cell types in the liver ($CD45^-$) including hepatocytes using flow cytometry. The sorted cells were mixed at the same proportion among different biological replicates and subjected to single-cell transcriptomic analysis. We determined the transcriptome of 11,085 cells from four biological replicates ($n = 2$ for both the sham-treated group and 4T1 breast cancer-bearing group, respectively). After processing the data, we found 23 clusters of cells (Supplementary Fig. 5a). Using known marker genes, we identified hepatocytes, hepatic stellate cells, endothelial cells, conventional dendritic cells, plasmacytoid dendritic cells, T cells, B cells, macrophages (including Kupffer cells), neutrophils, basophils, natural killer cells, and other monocyte-derived cells (Fig. 2f and Supplementary Fig. 5b; Supplementary Data 4). These scRNA-seq datasets allowed us to link the differentially expressed biological pathways with their underlying cell types, helping the interpretation of our spatial transcriptomics datasets.

Using these datasets, we examined which cell types accounted for the changes in spatial transcriptomics. Such comprehensive analyses resulted in the following findings: (i) breast cancers rewire zonated gene expression patterns in hepatocytes (Figs. 3 and 4) and (ii) breast cancers activate innate immune cells particularly neutrophils in distinct zonated manners (Fig. 5). These findings are discussed in the following two sections.

**Altered hepatocyte gene expression in zonated manners**. As shown in Fig. 2d, there was a general downregulation of genes related to metabolic processes in hepatocytes. This trend was validated using bulk RNA-seq datasets that we previously reported[8], suggesting that breast cancer transplantation repressed metabolism in the liver (Supplementary Fig. 6). As an example, *Cyp2a22* was down-regulated in the whole liver. Intriguingly, among the repressed processes, we observed a mixture of processes (and genes) that followed zonation patterns, and some that did not. For example, the zonation pattern of genes involved in "xenobiotic catabolic processes" appeared almost completely lost in cancer-bearing samples (Fig. 2d). In contrast, genes involved in "aspartate family amino acid metabolic processes" were generally downregulated in the cancer-bearing samples, yet continued to show a clear zonation pattern (higher expression in the $Alb^{high}$ zone; Fig. 2d).

We tried to further support these observations by visualization of Visium spots and UMAP plots. As exemplified by the xenobiotic metabolism pathway, the zonated pattern of this pathway seemed strongly disrupted by 4T1 breast cancers (Fig. 3a). This was evident in the corresponding UMAP plot, where the biased localization of this pathway towards $Cyp2e1^{high}$ spots was no longer detectable (Fig. 3b). Importantly, scRNA-seq datasets validated that cells enriched for this biological pathway were indeed hepatocytes (Fig. 3c). This was a typical pattern of pathways showing "loss of zonation" accompanied by general down-regulation of gene expression. On the other hand, despite being down-regulated, zonation remained relatively intact for "aspartate family amino acid metabolic processes" in $Alb^{high}$ zones (Fig. 3d–f). We observed a similar trend in other pathways such as the "triglyceride catabolic process" (Fig. 2d and Supplementary Fig. 7a–c). These results indicated that the effects

of breast cancers on liver zonation differ depending on biological pathways.

We also noted that the acute phase response was massively induced in hepatocytes in somewhat zonated manners (Fig. 4a). *Serum amyloid alpha* (*Saa*) 1 and 2 were representative of this observation (Supplementary Fig. 7d, e). These genes were expressed at a very low level in the sham livers but strongly induced by cancer transplantation. Interestingly, this induction occurred in a zonated manner: acute phase response genes were abundantly expressed in $Alb^{high}$ zones (Fig. 4b, c), and was driven predominantly by hepatocytes (Fig. 4d). This result together with the repressed metabolism in $Alb^{high}$ hepatocytes implicate hepatocytes in switching their focus from metabolism to acute phase response in the presence of breast cancers. These results indicated that breast cancers affected hepatocyte gene expression in zonated manners, which is possibly related to the disrupted liver metabolism observed in the bulk metabolome data we previously reported[4,8].

**Zonated immune cell activation in the liver**. The heatmap shown in Fig. 4a suggested that breast cancers induced immune cell activation and infiltration in the liver. Indeed, the most striking difference between the sham and the cancer-bearing scRNA-seq samples was the appearance of a spectrum of neutrophils in the cancer-bearing liver data (Fig. 5a and Supplementary Fig. 8). We validated the increase in neutrophils in the liver using flow cytometry (Fig. 5b). Interestingly, these neutrophils were positive for *Mki67*, a marker for cell proliferation (Fig. 5c). This implied that neutrophil proliferation might be enhanced in the bone marrow. To verify this possibility, we used MAgnetic Cell Sorting (MACS) to separate neutrophils and other cell types in the bone marrow, and quantified the expression of *Mki67* in both populations by qPCR. We found that *Mki67* was already high in bone marrow neutrophils, supporting our hypothesis (Fig. 5d). It remained unclear if neutrophils proliferated in the liver. We assumed that *Mki67* expression in neutrophils in the liver might be remnants of neutrophil proliferation in the bone marrow. Moreover, the neutrophil proportion was elevated in the bone marrow (Fig. 5e). RNA velocity analysis using velocyto.R[29] suggested the existence of a flow between several of the neutrophils clusters (Fig. 5f), in particular from clusters that are exclusively observed in the cancer-bearing samples towards the cluster observed almost exclusively in the sham samples. Interestingly, this flow was correlated with the expression of various neutrophil marker genes such as *Cxcr2* (Supplementary Fig. 9): *Cxcr2* was strongly expressed in the sham subcluster of neutrophils (i.e., normal mature neutrophils). Other genes such as *Mpo* were strongly expressed in the cancer-specific clusters that were enriched for *Mki67*. These results led to the hypothesis that neutrophils immigrating into the liver of cancer-bearing mice are predominantly immature, differentiating neutrophils, whereas neutrophils in the livers of sham mice are mature neutrophils. These immature neutrophils might represent myeloid-derived suppressor cells (i.e., pathologically activated neutrophils and monocytes)[30,31].

Our data suggested a biased (i.e., non-uniform) localization of these neutrophils in the livers of 4T1 cancer-bearing mice. Although the zonation pattern was hard to spot by eye in the Visium slices, the pattern was clear in the UMAP plot, showing especially high signals in the $Alb^{inter}$ zone of the cancer-bearing liver (Fig. 5g–i). This pattern was also seen in other immune-related pathways as shown in Fig. 4a. These results suggested an influx of neutrophils into the livers of cancer-bearing animals, preferentially to regions of the liver that are not high in *Alb* expression, but rather between the portal and central veins.

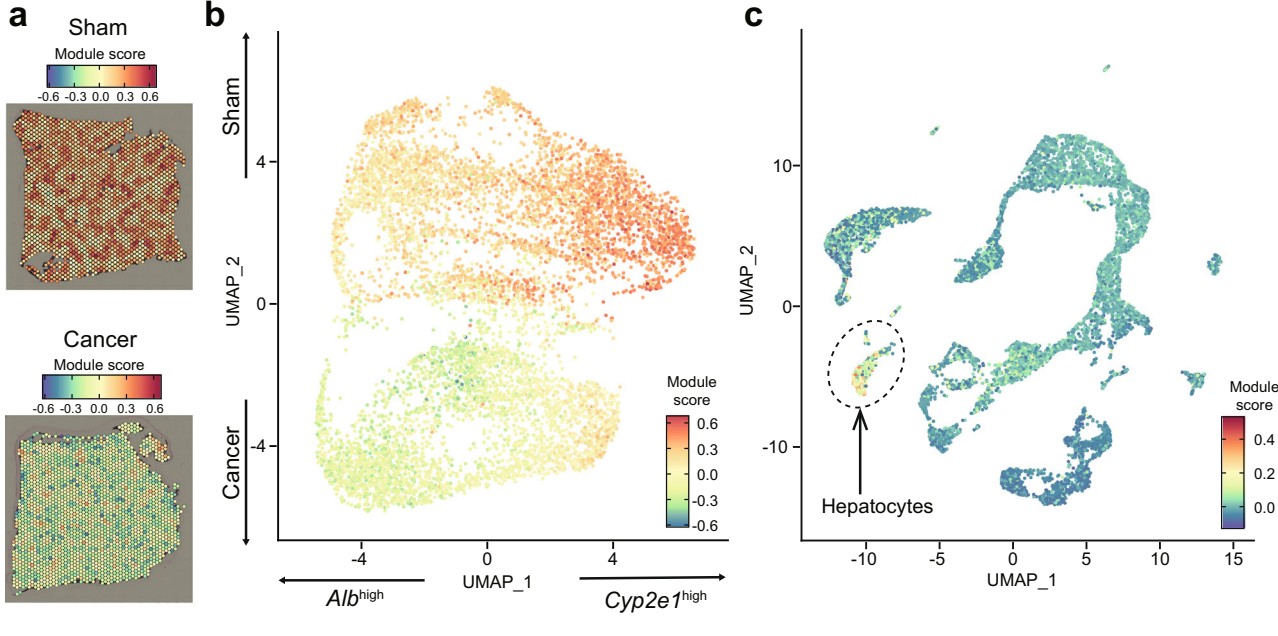

Xenobiotic catabolic process

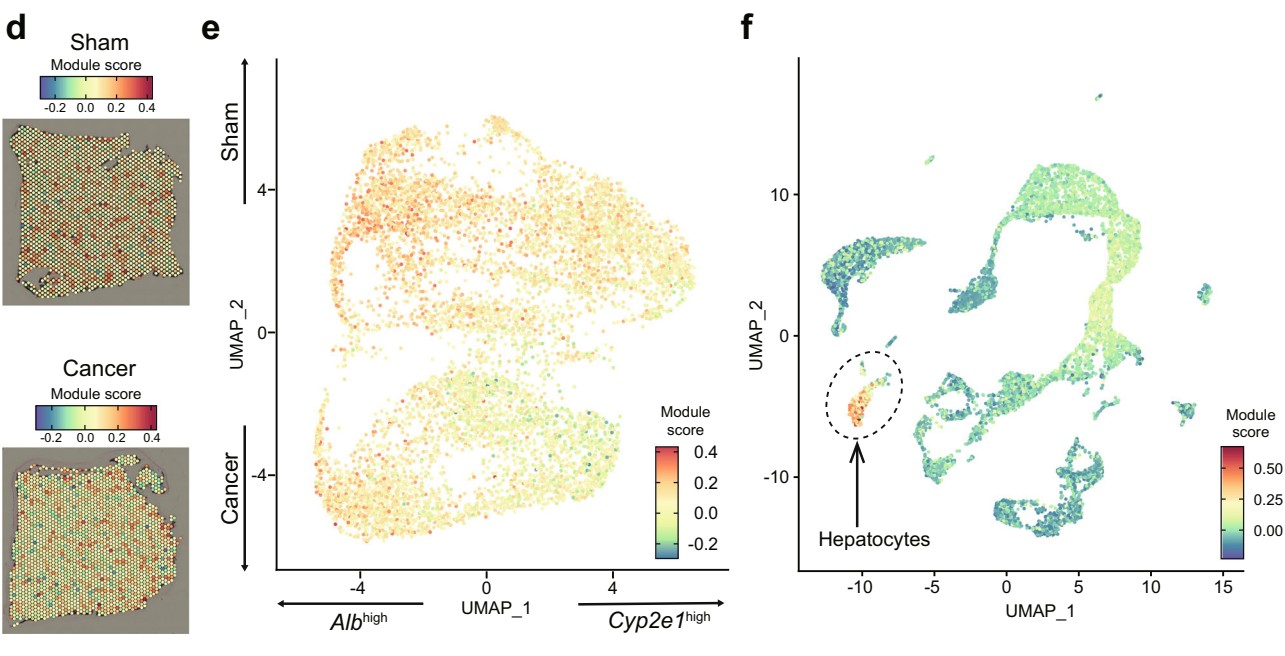

Aspartate family amino acid metabolic process

**Fig. 3 4T1 breast cancers disrupt liver zonation in various manners. a–c** For the genes associated with the GO term "xenobiotic catabolic process". **a** Module scores in one of the sham and cancer Visium samples. **b** The same module scores in a UMAP representation of the Visium data. **c** Module scores in the scRNA-seq data, showing high scores predominantly in the hepatocyte cluster. **d–f** Same as **a–c** for genes associated with the "aspartate family amino acid metabolic process" GO term.

Indeed, several marker genes of neutrophils had particularly high expression levels at this subset of spots (Supplementary Fig. 10). These data were in line with the heatmap showing the activation of the immune system in zonated manners (Fig. 4a; see for example "neutrophil activation involved in immune response"), suggesting that neutrophil distribution in the livers of cancer-bearing mice was not uniform.

Although we initially focused on neutrophils, we also found the activation of other cell types. We found that macrophages were activated in zonated manners, most likely representing activation of liver resident macrophages (i.e., Kupffer cells) (Supplementary Fig. 11a–c). Macrophage spots were observed from the $Alb^{inter}$ to $Alb^{high}$ zones, a pattern distinguishable from neutrophils (Fig. 5h). The increase of basophils was also notable in scRNA-seq and flow cytometry (Fig. 2f and Supplementary Fig. 11d, e). Basophils were not present in the sham controls in the scRNA-seq datasets (Supplementary Fig. 8). Using flow cytometry, we detected an increase in basophils in the liver whereas they were reduced in the bone marrow.

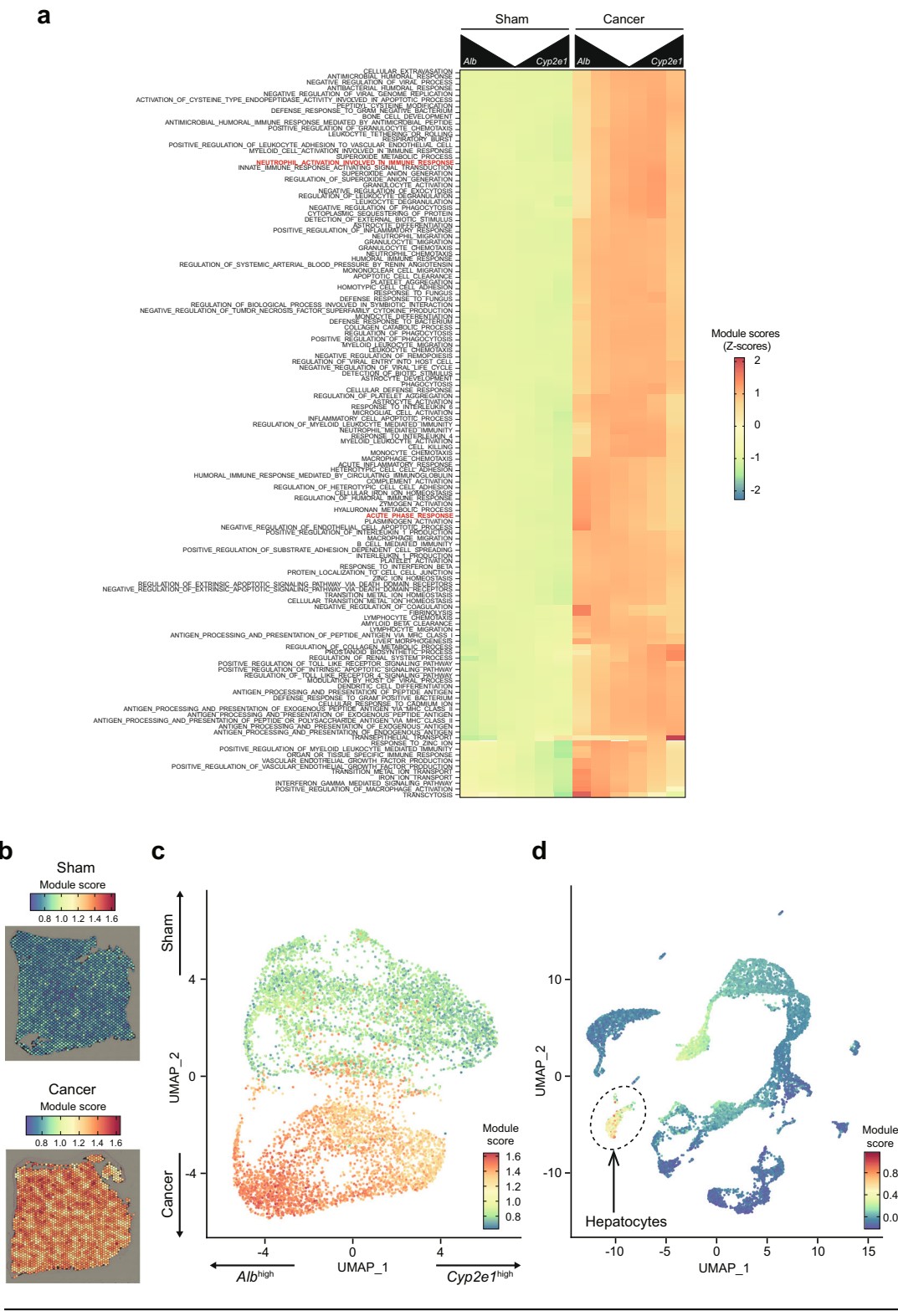

**Fig. 4 4T1 breast cancers induce acute phase response in a zonated manner. a** Heatmap showing the normalized module scores of selected GO terms in the function of their distance to *Alb*^high^ and *Cyp2e1*^high^ spots in sham (left) and cancer-bearing (right) samples. **b**, **c** Genes involved in the acute phase response are induced in the liver of cancer-bearing mice. **b** Module scores in one of the sham and cancer Visium samples. **c** The same module scores in a UMAP representation of the Visium data. **d** Module scores in the scRNA-seq data, showing high scores predominantly in the hepatocyte cluster.

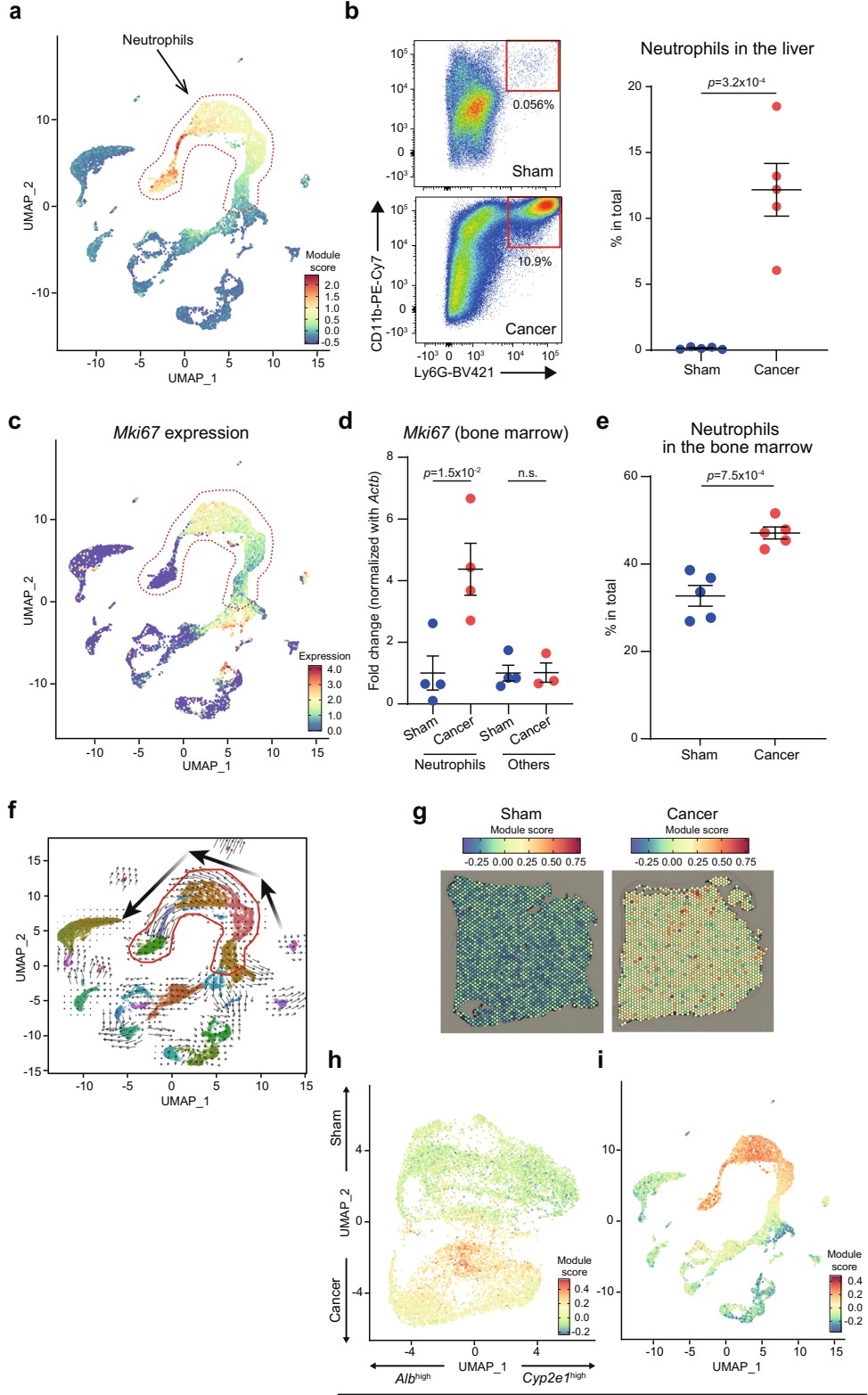

**Neutrophil activation involved in immune response**

These results suggested the enhanced influx of basophils from the bone marrow to the liver (Supplementary Fig. 11d, e). Unlike neutrophils, it appeared that basophils exhibited neither in situ differentiation nor zonated localization. Given their roles in expressing various unique cytokines, these innate immune cells might contribute to the altered spatial transcriptomics landscape of the livers in various distinct manners.

**Alterations in the transcytosis pathway**. Our data revealed other zonated patterns of gene expression in the livers of cancer-bearing mice whose significance is relatively unclear to us at present (Supplementary Fig. 12). The transcytosis pathway was much more active in $Alb^{high}$ zones rather than in $Cyp2e1^{high}$ zones (Fig. 4a and Supplementary Fig. 12a–c). 4T1 cancer transplantation elevated the expression of genes involved in this pathway,

**Fig. 5 Zonated immune cell activation in the livers of cancer-bearing mice. a** UMAP plot of the scRNA-seq data showing the module score of neutrophil marker genes. **b** Flow cytometric analysis of Ly6G$^+$CD11b$^+$ neutrophils in the livers of sham and 4T1-bearing mice. Representative plots are shown on the left. Data are represented as the mean ± SEM. The *p*-value is shown (unpaired two-tailed Student's *t*-test). *n* = 5. **c** UMAP plot of the scRNA-seq data showing the expression of *Mki67*. **d** Expression of *Mki67* in the bone marrow of sham and 4T1-bearing mice. Neutrophils and non-neutrophils are MACS-sorted and analyzed by qPCR. Data are represented as the mean ± SEM. The *p*-value is shown (unpaired two-tailed Student's *t*-test). *n* = 4 for neutrophils and other cell types in sham, and neutrophils in 4T1-bearing mice. *n* = 3 for other cell types in 4T1-bearing mice. **e** Flow cytometric analysis of Ly6G$^+$CD11b$^+$ neutrophils in the bone marrow of sham and 4T1-bearing mice. Data are represented as the mean ± SEM. The *p*-value is shown (unpaired two-tailed Student's *t*-test). *n* = 5. **f** UMAP plot of the scRNA-seq data showing the predicted dynamics of cells using RNA Velocity. The three big arrows indicate the dynamics of immature neutrophils toward mature neutrophils. **g–i** The module scores of genes associated with GO term "neutrophil activation involved in the immune response" in the Visium slices (**g**) and in a UMAP plot of the same data (**h**). **i** Module scores in the scRNA-seq data, showing high scores predominantly in the neutrophil populations.

potentially enhancing transcytosis in endothelial cells nearby portal veins. In an interesting contrast, the transepithelial transport appeared zonated towards *Cyp2e1*$^{high}$ zones and activated in the presence of 4T1 breast cancers (Fig. 4a and Supplementary Fig. 12d–f). A previous study reported immune cell zonation orchestrated by liver sinusoidal endothelial cells[23]. In this report, the authors suggested that liver sinusoidal endothelial cells sense the microbiome, accordingly modulating immune cell localization. The complex interaction among different cell types and stimuli might underlie the altered zonation of immune cells and endothelial cells in our experimental settings. Addressing these zonation patterns in detail is also an important next step.

It is also important to clarify the involvement of liver extrinsic factors in the disruption of liver zonation. It is known that cancers cause anemia[32], which might be an extrinsic factor. To test if cancers cause anemia phenotypes in our experimental settings, we analyzed the bone marrow of sham and 4T1-bearing mice. We found that 4T1 transplantation induces various gene expression changes in the bone marrow (Supplementary Fig. 13a). Gene ontology analyses and GSEA suggested the downregulation of erythrocyte differentiation and heme metabolism (Supplementary Fig. 13b, c). Bone marrow looked red in the sham group. Such red color became relatively faint in the 4T1 group, suggesting that erythrocytes decreased upon 4T1 transplantation (Supplementary Fig. 13d, e). On the basis of these, we concluded that 4T1-bearing mice likely suffer from anemia, which might be an important extrinsic factor in the disruption of liver zonation.

Together, our results suggest that the induced gene expression in cancer-bearing mice could be attributed to at least two different processes; the influx of immune cells (especially neutrophils) and the response of liver-resident cells (especially hepatocytes) to distal cancer and/or the influx of immune cells.

## Discussion

Although it is known that solid cancers remotely disrupt liver homeostasis, it has been unclear whether this disruption accompanies abnormalities in liver zonation. In the current study, we addressed this question by combining spatial transcriptomics and single-cell transcriptomics, finding that a murine breast cancer model disorganizes liver transcriptome in zonated manners.

Our analyses integrating spatial transcriptomics and single-cell transcriptomics allowed us to categorize breast cancer-induced gene expression changes in the bulk livers according to liver zonation. We found that some gene expression changes were coincident with loss of zonation, as exemplified by the xenobiotic catabolic process in the *Cyp2a1*$^{high}$ zone (Fig. 3a–c). On the other hand, zonation of the aspartate family amino acid metabolism and the triglyceride catabolic process was relatively intact in the *Alb*$^{high}$ zone (Fig. 3d–f and Supplementary Fig. 7a–c). In addition, acute phase protein genes *Saa1* and *Saa2* showed an intriguing pattern of activation: their activation was not uniform within the liver. Rather, *Saa1* and *Saa2* were induced in the *Alb*$^{high}$ zone

(i.e., gain of zonation), suggesting a zonation-distinct gene expression mechanism underlying acute phase response (Fig. 4 and Supplementary Fig. 7d, e). It is likely that *Saa* expression is biased toward secreting hepatocytes, which is consistent with zonated protein secretion in the portal layers[14,33]. These results indicate that the effects of breast cancers on liver zonation were distinct according to biological pathways, revealing previously unrecognized complexity of cancer-induced gene expression changes in the liver. It remains unclear how these pathway-specific changes in zonation occurred. Further detailed investigation addressing this issue is critical to deepening our knowledge of how solid cancers and other diseases disrupt liver homeostasis.

The increased influx of immune cells to various peripheral organs is a hallmark of host pathophysiology in cancers[1,10,34]. This is also the case for the liver, and many studies pointed out the massive infiltration and activation of particularly innate immune cells in the cancer-bearing condition[2–5,8,34]. Our data detected various zonated patterns of immune cell activation in the liver (Fig. 5 and Supplementary Figs. 10, 11). We found that potentially immature neutrophils infiltrated the liver in the *Alb*$^{inter}$ zone. It seems that they were gradually changing their transcriptome status within the liver, possibly implicating differentiation in situ (Fig. 5f). We also captured the zonated activation of liver-resident macrophages without such in situ transcriptomic changes (Supplementary Fig. 11a–c). Basophils followed neither the neutrophil pattern nor the macrophage pattern (Supplementary Fig. 11d, e). It appeared that basophils migrated from the bone marrow to the liver, forming a single cluster. Collectively, our analyses revealed intriguing, distinct patterns of influx and activation of various innate immune cells to the liver, which we assume are important to understand the basis of the altered liver zonation and disrupted liver homeostasis in the cancer-bearing condition. How these unique patterns are generated and how such altered immune cell dynamics affect other cell types especially hepatocytes are important questions to be answered.

Our study is currently limited to a murine breast cancer model, but a disease-dependent disruption in liver zonation could be general[15]. For example, pathogenesis such as non-alcoholic fatty liver disease (NAFLD), non-alcoholic steatohepatitis (NASH), and liver cirrhosis develop in a somewhat zonated manner. In detail, these develop with steatosis and inflammation in the pericentral region[35,36]. In an intriguing contrast, various liver injuries begin periportally[37–39]. These indicate that each liver pathology has a distinct pattern of disrupted liver zonation, which possibly underlies the nature of such pathology. Obtaining spatial transcriptomics datasets from various disease conditions will be useful to advance our understanding of various liver diseases in light of alterations in zonated gene expression.

Given the nature of cancer transplantation models, disease progression in our experimental settings could be considered extreme. Compared to other pathologies described above, which

initiate locally in the specific region of the liver, the altered zonation in our model was consistent throughout the spatial transcriptome sections. These results led to an interpretation that the effects of cancer transplantation on liver zonation were strong, reflecting the terminal state of diseases. We imagine that disruption in zonated gene expression might begin in more region-specific manners in the livers of, e.g., metastatic cancer patients. The lack of human data is a limitation in our study, and future investigation of liver zonation in human cancer patients is critical to address the hypothesis we described. Yet, we believe that the current datasets from a murine breast cancer model will be a basis for gaining insights into the terminal state of liver abnormalities in the cancer-bearing condition.

Cancers cause various systemic effects on the host[1-11]. These alterations likely interact with each other. For example, anemia could be an extrinsic factor for the disruption of liver zonation (Supplementary Fig. 13). On the other hand, the massive down-regulation of e.g., triglyceride metabolism should affect non-hepatocyte cell types within and outside the liver, potentially altering the immune system status. These abnormalities of the host might benefit cancer growth. Dissecting such complex interactions at the body-wide level is critical to understand cancer's adverse effects on the host.

In summary, we demonstrate uniquely altered patterns of zonated gene expression in the liver. Our study highlights the strengths of the combination of spatial transcriptomics and single-cell transcriptomics to understand cancer's adverse effects on the liver and to reveal complex interactions among various cell types in disease conditions.

## Methods

**Mice**. All animal experiment protocols were approved by the Animal Care and Use Committee of Kyoto University. Wild-type BALB/c female mice (8-week-old) purchased from Japan SLC Inc. (Hamamatsu, Japan) were housed in a 12-h light/dark paradigm with food (CE-2, CLEA Japan, Inc., Tokyo) and water available ad libitum. No blinding was done in the experiments described in this study. These were reported in accordance with the *Animal Research: Reporting In Vivo Experiments* (ARRIVE) guidelines.

**Cell line**. 4T1 mouse breast cancer cell line[8] was cultured in RPMI1640 (nacalai tesque, Kyoto, Japan) supplemented with 10% fetal bovine serum (nacalai tesque) and 1% penicillin/streptomycin (nacalai tesque) in a 5% $CO_2$ tissue culture incubator at 37 °C. This cell line was originally a gift from Dr. T. Nojiri and has been cultured in our group for many years. 4T1 cells were tested for mycoplasma contamination before transplantation using MycoFlour™ Mycoplasma Detection Kit (Thermo Fisher Scientific, MA, USA). This cell line was authenticated by RNA-seq and histology.

**Cancer transplantation**. $2.5 \times 10^6$ 4T1 cells in 100 µl of RPMI1640 medium or the same amount of cell-free RPMI1640 medium (sham) were inoculated subcutaneously into the right flank of 8–10-week-old BALB/c females and these mice were sacrificed 14 days after transplantation. The confluency of 4T1 cells was ~70–80% prior to injection. The formation of primary cancer tissues was confirmed by eye upon sacrificing mice.

**Spatial transcriptomics**. The livers from sham and 4T1-bearing mice were embedded in OCT Compound (Sakura Finetek Japan, Tokyo, Japan) and frozen samples embedded in OCT compound were sectioned at a thickness of 10 µm (Leica CM3050S). We utilized the 10x Genomics Visium Spatial Gene Expression Solution for spatial transcriptomics. Libraries for Visium were prepared according to the Visium Spatial Gene Expression User Guide (catalog no.CG000239Rev A, 10x Genomics). Mouse livers were mounted for 4 stages (two sham samples and two 4T1 cancer-bearing samples) with one stage per capture area on a 10x Visium gene expression slide containing four capture areas with 5000 barcoded RNA spots printed per capture area. Spatially barcoded cDNA libraries were built using the 10x Genomics Visium Spatial Gene Expression 3′ Library Construction V1 Kit. The optimal permeabilization time for 10 µm thick liver sections was determined to be 6 minutes using 10x Genomics Visium Tissue Optimization Kit (catalog no. CG000238 Rev A, 10x Genomics). H&E staining of the liver tissue sections was imaged using a light microscope (Keyence BZ-X800, Keyence Ltd. HQ & Laboratories, Osaka, Japan), and the images were stitched and processed using Keyence BZ-X800 Analyzer software. The library was sequenced using the NovaSeq

6000 system (Illumina) according to the manufacturer's instructions. Sequencing was carried out using a 28/90 bp paired-end configuration, which is sufficient to align confidentially to the transcriptome. Raw FASTQ files and histology images aligned to the mouse reference genome provided by 10x Genomics (mm10 Reference-2020-A) using the Space Ranger 1.2.1 pipeline to derive a feature spot-barcode expression matrix (10x Genomics).

### Spatial transcriptomics data analysis

*Data processing.* Spatial transcriptome data, including UMI counts and spot coordinates, were analyzed using the R Seurat package (version 4.0.0)[40] The four images and their data were processed into Seurat objects using the Load10X_-Spatial function and merged into a single Seurat object. This resulted in a dataset of 7758 spots and 32,285 genes. Data were processed using a standard Seurat work-flow (SCTransform function). We conducted batch effect reduction between the four slices using the SelectIntegrationFeatures, PrepSCTIntegration, FindInte-grationAnchors, and IntegrateData functions, followed by dimensionality reduction using PCA and UMAP (based on 30 principal components).

*Prediction of spatially variable genes.* We predicted spatially variable genes (SVGs) using singleCellHaystack (version 0.3.4)[27]. The haystack_highD function detects genes whose expression is distributed in a non-random pattern inside the input space (=here the 2D spatial coordinates of the spots). For each of the 4 slices, the inputs to the haystack_highD function were 1) the spatial coordinates of the spots and 2) the detection data of each gene at each spot. Genes were defined as "detected" if their signal in a spot was higher than the median signal in the sample. Additional input parameters to haystack_highD were grid.points = 500, and scale = FALSE. To avoid biases by damaged cells, spots with a high proportion of mitochondrial UMIs (top 10%) were excluded from this analysis.

*Analysis of zonation of genes sets and their module scores.* Using the R package msigdbr (version 7.5.1) we defined sets of genes associated with Gene Ontology (GO) biological process functional annotations. After filtering out GO terms associated with <20 or >250 genes present in our data, we retained 2898 GO terms and their associated genes (Supplementary Data 1). For each of these sets, we calculated module scores using Seurat's AddModuleScore function in both the spots of the spatial transcriptomics data as well as in the individual cells of our scRNA-seq data[40]. In brief, modules scores reflect the average expression of the set of genes within each spot or cell, subtracted by the average expression of a number of control genes with similar average expression levels as the input set. AddMo-duleScore was used using default parameters; all genes were divided into 24 bins based on their expression level (nbin = 24) and for each input gene 100 genes were randomly selected from the same bin (ctrl = 100) to serve as a control set.

To characterize zonation in the patterns of activity of gene sets, we divided the Visium spots of each image into three sets of equal size: those with the highest *Alb* expression (*Alb*^high), those with the highest *Cyp2e1* expression (*Cyp2e1*^high), and the remaining one-third (with intermediate *Alb* and *Cyp2e1* expression. Next, we compared the module scores of each GO term in the *Alb*^high zone versus the *Cyp2e1*^high zone using the Wilcoxon Rank Sum test (R function wilcox.test). Differences in the mean module scores of the *Alb*^high zone versus the *Cyp2e1*^high zone (X-axis) and the p-value of the Wilcoxon Rank Sum test (log₁₀; Y-axis) were visualized using a volcano plot (Fig. 1b).

To characterize the zonation patterns at a higher resolution, we first defined *Alb*^high spots as the top 10% of spots with the highest *Alb* expression in each image. In the Visium platform, spots are organized in a hexagonal pattern. Each spot has six closest neighbors (at a distance of 100 µm center-to-center), and six secondary neighbors (at a distance of about 173 µm center-to-center). For each image, we defined *Cyp2e1*^high spots and their closest and secondary neighbors in the same way. Next, for each GO term, we calculated the mean of the module scores in the *Alb*^high and *Cyp2e1*^high spots and their neighbors (Supplementary Data 3). The 200 GO terms with the largest ranges of values were picked up for visualization using heatmaps (Figs. 2d and 4a).

**Single-cell RNA-seq**. The livers of sham and 4T1-bearing mice were minced and dissociated in RPMI1640 containing DNase I (100 µg/ml: Roche, Switzerland) and collagenase IV (1 mg/ml: Worthington Biochemical Corporation, NJ, USA). Red blood cells (RBCs) were lysed with RBC lysis buffer (BD Bioscience, NJ, USA). The obtained cells were stained with anti-CD45 antibody (Clone: 30-F11 (BVD421)), anti-CD19 antibody (Clone: 1D3), and anti-TCRβ antibody (Clone: H57-597) for 30 min at 4 °C. Then we sorted and mixed 20,000 CD45⁻ cells that include hepa-tocytes, 6000 CD45⁺CD19⁺TCRβ⁻ cells (i.e., B cells), 6000 CD45⁺CD19⁻TCRβ⁺ cells (i.e., T cells), and 20,000 CD45⁺CD19⁻TCRβ⁻ cells (e.g., neutrophils) using FACS Aria (BD Bioscience, NJ, USA). The obtained mixture of cells was subjected to single-cell transcriptomics.

The single-cell RNA-seq library was constructed by using the Chromium Controller and Chromium Single Cell 3′ Reagent Kits v3.1 (10x Genomics) following the standard manufacturer's protocols (catalog no. CG000227 Rev C). Cells sorted from fresh live mouse liver cells were washed with PBS, then pipetted through a 40-µm filter to remove cell doublets and contamination. Cell viability was confirmed by trypan blue staining. The collected single-cell suspension was

immediately loaded onto the 10x Chromium controller to recover 10,000 cells, followed by library construction. The library was sequenced using the NovaSeq 6000 system (Illumina). Sequencing was carried out using a 28/90 bp paired-end configuration. After sequencing analysis, the obtained fastq files were mapped to the reference genome provided by 10x Genomics (refdata-cellranger-mm10-3.0.0) and quantified read count using the Cell Ranger ver 3.1.0 count pipeline (10x Genomics) with the default parameters.

**Single-cell RNA-seq data analysis**. The UMI counts for the four samples were analyzed using the R Seurat package (version 4.0.0)[40]. Initial exploratory analysis revealed that one of the sham samples contained a large population of cells with a relatively high fraction of mitochondrial reads, which formed a separate cluster of cells after dimensionality reduction. To avoid biases caused by these cells, we applied a more stringent filter for this sample, removing cells with more than 5% mitochondrial UMIs. For the remaining three samples, we removed cells with more than 20% mitochondrial UMIs. In addition, for all samples, we removed cells with fewer than 200 detected genes. This resulted in a dataset with 11,085 cells (sham 1: 364; sham 2: 2660; cancer-bearing 1: 4769; cancer-bearing 2: 3302) and 19,355 genes.

Data were processed and subjected to dimensionality reduction using a standard Seurat workflow, including normalized, selection of 2000 variable features, scaling, principal component analysis (PCA), and UMAP using the first 20 PCs. The clustering of cells was done using Seurat's FindNeighbors function (using 20 PCs) and FindClusters function (using resolution 0.5), resulting in 23 clusters.

We assigned a cell type to each cluster after inspection of the expression patterns of a selection of known marker genes (Supplementary Data 4).

RNA Velocity analysis was performed using velocyto.py (version 0.17.17), and the R packages velocyto.R (version 0.6), and SeuratWrappers (version 0.3.0) (https://github.com/satijalab/seurat-wrappers)[29]. First, we ran velocyto.py on each of the 4 scRNA-seq samples using option run10x, the mm10 gene annotation file, and a repeat annotation file (option -m) obtained from the UCSC Genome Browser[41]. Resulting.loom files were processed and visualized using functions ReadVelocity, RunVelocity, and show.velocity.on.embedding.cor.

**Bulk liver RNA-seq data analysis**. We used DESeq2 (version 1.28.1) to normalize the read count per gene for all 8 liver samples (4 sham and 4 cancer-bearing from DRA01192)[8,42]. Differentially expressed genes (DEGs) between sham and cancer-bearing samples were predicted using default parameters.

**Flow cytometry**. The livers and bone marrow cells were harvested from sham and 4T1 cancer-bearing animals 14 days after transplantation. Red blood cells (RBCs) were lysed with RBC lysis buffer (BD Bioscience). The obtained cells were then stained with the following antibodies (BioLegend, CA, USA) for 30 min at 4 °C: anti-CD45 antibody (Clone: 30-F11), anti-CD19 antibody (Clone: 6D5), anti-TCRβ antibody (Clone: H57-597), anti-CD11b antibody (Clone: M1/70), anti-Ly6G antibody (Clone: 1A8), anti-FCεR1a antibody (Clone: MAR-1), anti-CD117 antibody (Clone: 2B8), anti-CD123 antibody (Clone: 5B11), anti-CD49b antibody (Clone: DX5), anti-CD3 antibody (Clone: 17A2), anti-CD11c antibody (Clone: N418), anti-TER-119 antibody, or anti-CD34 antibody (Clone: HM34). Non-viable cells were stained with Propidium Iodide Solution (PI) and gated out. Data were acquired using FACS Aria (BD Bioscience) and analyzed using FlowJo software (v10.8.1, BD Biosciences). Gating strategy is shown in Supplementary Fig. 14.

**Bone marrow collection**. For MACS-based neutrophil isolation and flow cytometry, the hind limb femur and tibia from both legs of sham and 4T1-bearing mice were collected in RPMI1640 media (nacalai tesque) in a 60 mm dish. The femur and tibia were transferred to a new dish with RPMI1640 medium after associated muscles and connective tissues were removed. The femur and tibia were cut and perfused using a syringe needle. The suspension containing immune cells was transferred to a clean tube and subjected to MACS-based neutrophil isolation and flow cytometry.

For bone marrow RNA-seq experiments, the bone marrow was collected according to the previous publications[43,44]. The right femurs were collected, cleaned of muscle tissue, and then polished with gauze. The femurs were placed with knee-end down in a perforated 0.6 ml tube added 80 μl of RNAlater (Qiagen) and inserted in a 1.5 ml centrifuge tube, followed by centrifugation at 5700 × g for 30 s. The resulting pellets were suspended in 1 ml of Trizol reagent (Thermo Fisher Scientific) and proceeded with RNA extraction.

**MACS**. Neutrophils and the other cell types were isolated from the suspension using the MACS Neutrophil isolation kit (Miltenyi Biotec, 130-097-658) according to the manufacturer's instructions. The obtained cells were then subjected to RNA isolation, cDNA synthesis, and qPCR analysis.

**RNA extraction, cDNA synthesis, and qRT-PCR**. The collected bone marrow cells were homogenized with Trizol reagent (Thermo Fisher Scientific). Total RNA was extracted from the homogenized supernatant using RNeasy Mini Kit (Qiagen, Venlo, Netherlands) according to the manufacturer's instruction and reverse-transcribed with the aid of Transcriptor First Strand cDNA synthesis kit (Roche,

Switzerland). qPCR experiments were performed using the StepOnePlus qPCR system (Applied Biosystems, CA, USA) and SYBR Green Master Mix (Roche, Basal, Switzerland). *Actb* was used as an internal control. The primers used in these experiments are:

*Mki67* (Forward: AAAAGTAAGTGTGCCTGCCC,
Reverse: GGAAAGTACGGAGCCTGTAT),
*Actb* (Forward: CGGTTCCGATGCCCTGAGGCTCTT,
Reverse: CGTCACACTTCATGATGGAATTGA)
Quality check data for our qPCR are provided in Supplementary Data 5.

**Bone marrow RNA-seq analysis**. Total RNAs were extracted from bone marrow cells as described above. The RNA quality was assessed by Agilent 2100 bioanalyzer using the RNA 6000 Nano Chip obtained from Agilent Technologies (Amstelveen, The Netherlands). RNA-seq libraries were generated using the Illumina TruSeq Stranded mRNA preparation kit according to the manufacturer's instructions (Illumina, CA, USA). Sequencing experiments were performed with NovaSeq 6000 system (Illumina; Single End 100 bp). After quality control of the data using Fastp tool version 0.20.0[45], trimmed reads were mapped to mm10 by STAR version ver2.6.0a[46]. Read counts were normalized with the reads per million per kilobase (RPKM) method. The generated gene expression matrix with RPKM scores is listed in Supplementary Data 6. DEGs were obtained using DEseq2 with default parameters[42]. Gene expression matrix was used to perform gene set enrichment analysis (GSEA) to interpret transcriptional profiles[47,48]. Enrichment score (ES), normalized enrichment score (NES), and false discovery rate (FDR) for all variables and signatures were obtained. Gene ontology (GO) analyses were performed using an online tool DAVID (https://david.ncifcrf.gov/). The bubble plot was produced using ggplot2 (https://ggplot2.tidyverse.org/index.html) to visualize DEGs and enriched pathways.

**Statistics and reproducibility**. Sample sizes were empirically determined. In scRNA-seq and spatial transcriptomics experiments, we acquired data from two mice in each experimental group. In other experiments, $n = 3$-5 was set as a threshold. Significant differences between the two groups were estimated using two-tailed, unpaired Student's *t*-tests. Statistical differences in modules scores were evaluated using the Wilcoxon Rank Sum test as described in the relevant section of the "Methods" section. Data can be explored in the Broad Institute Single Cell Portal under accession numbers SCP2045 (scRNA-seq) and SCP2046 (Visium). This includes the raw count data as well as the processed data, the Seurat objects, UMAP embeddings, and zone information for the Visium data.

**Reporting summary**. Further information on research design is available in the Nature Portfolio Reporting Summary linked to this article.

## Data availability
The Visium and scRNA-seq datasets in this study are available in DNA Databank of Japan (DDBJ) under the accession numbers of DRA014802 and DRA009332, respectively. These data are also available in the Broad Institute Single Cell Portal under accession numbers SCP2045 (scRNA-seq) and SCP2046 (Visium). The bulk transcriptome data from the liver and bone marrow are also available (DRA01192[8] and DRA015216, respectively). Source data file is provided as Supplementary Data 7.

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

## Acknowledgements

We thank Dr. Diego Diez (Osaka University) and Prof. Kosuke Yusa (Kyoto University) for their helpful discussions and advice. This work was supported by JSPS KAKENHI (16H06279, 18K15409, 18H04810, 20H03451, 20H04842, and 22H04925; S.K., and 20K06609; A.V.), JST FOREST (JPMJFR2062; S.K.), JST Moonshot (JPMJMS2011-61; S.K.), Caravel, Co., Ltd (S.K.), Ono Medical Research Foundation (S.K), Takeda Science Foundation, The Uehara memorial foundation (S.K.), Chubei Ito foundation (S.K.) and Japan Foundation for Applied Enzymology (S.K). We thank Yuri Kawaoka for drawing the cartoons in Fig. 2e.

## Author contributions

A.V. performed and supervised data analyses, constructed figures, and wrote the manuscript. R.M. and R.K. performed in vivo experiments, analyzed data, and constructed figures. M.O. performed sequencing data acquisition, analyzed data, and made a substantial contribution to the discussion regarding the relevance of our work to human cancers. K.M., Y.K., and H.K. supported in obtaining Visium slices and flow cytometry analyses. A.S supported initial data analyses. C.H., Y.N., K.K., and M.T. contributed to the bone marrow analyses. M.S. and Y.T. made substantial contribution to the conception of this work. Y.S. supervised Visium and scRNA-seq experiments and data analysis. S.K. conceived and supervised this study, managed the collaboration in this study, analyzed data, and wrote the manuscript.

## Competing interests

The authors declare no competing interests.
