## [Peer Review File · Communications Biology]

Reviewers' comments:

Reviewer #1 (Remarks to the Author):

This manuscript tested the hypothesis that distant tumors affect liver zonation. To accomplish this, breast cancer-bearing female mice were used following injection of 4T1 breast cancer cells. The authors identified a zonal hepatocyte trajectory by identifying spatial RNAseq data rich in Alb expression (denoting periportal hepatocytes) and Cyp2e1 expression (denoting pericentral hepatocytes). Hepatocytes were also resolved into treatment groups: sham-operated and cancer bearing. Results suggest gene expression for certain biological pathways, such as triglyceride metabolism and aspartate biosynthesis were not zonally disrupted, while other pathways, such as those associated with xenobiotic metabolism, completely lose zonal distribution. Results from this study also highlight an influx in neutrophil infiltration in cancer bearing mice, with these infiltrating neutrophil populations showing gene expression signs of being immature. Taken together, the present study highlights cell-specific consequences in livers of 4T1 breast cancer bearing mice. The manuscript provides rich datasets that are thoroughly explored as well as presents novel hepatic cell-specific consequences due to breast cancer. The manuscript requires minor revisions to improve clarity and reproducibility.

Comments:

- The title would benefit from being more accurate to the work done such as using "Murine breast cancer cells disorganize ..." as the authors do not state whether solid tumors have formed. Breast cancer is often used instead of breast cancer in the manuscript. Is this truly a breast cancer model?
- The materials and methods section should include:
 - o Mouse strain details. Not provided when the model is first introduced and the sample size of treatment groups is not described. It is recommended that the authors review the ARRIVE guidelines (<https://arriveguidelines.org/>) and update the methods accordingly.
 - o It is unclear what the sham treatment is. Is it non-cancerous cells or just the vehicle or no manipulation?
 - o What was the confluency of the cultured cells prior to injection?
 - o What were the revision versions for the 10X Genomics user guides cited throughout the methods. For reproducibility 10X Genomics numbers their revisions typically as CG[number].
 - o Can the authors provide the list of filtered GO terms as supplemental material or using a data repository (e.g., Dataverse) for reproducibility? This will also assist readers in knowing that genes were considered in calculating pathway scores.
 - o How were the pathway scores calculated? It is only stated that they were calculated and differences were determined by WRS test.
 - o There is an overall lack of information on the bulk RNAseq. How was RNA isolated? was the RNA QC'd for RIN value? how were libraries prepared? what were the sequencing parameters? specify the reference genome?
 - o No QC metric provided for qPCR RNA input.
- The authors do not describe whether administration of cancer cells produced the anticipated phenotype. Did tumors develop and grow as expected in this model?
- The following figures need to be revised for clarity and interpretation.
 - o In the figures where the Visium slides are shown the size is so small that they look like dark squares. It may be more useful to move some of the figures as supplemental for easier visualization. The authors should also strongly consider uploading both the spatial and single-cell/nuclei data to the Broad Single Cell Portal for a more interactive visualization and provide the link in this publication. Several examples are available on how to deposit both types of data.
 - o Several figures are missing the label for interpretation or use a generic "value" term which makes it very difficult to interpret. Please use appropriate labels such as "activity score" instead of "value" and denote what the units represent for intensity in the Visium figures.
 - o Most of the Visium plots appear to be mislabeled as showing the expression of tmp1 which in addition to not being correct, doesn't follow normal mouse nomenclature (should be capitalized and italicized).
 - o Have the authors examined alternative means to plot spatial expression as a continuum across bins to improve representation of spatially resolved expression?
- The authors conclude identification of zonally resolved activated macrophages citing

supplementary Figure 10a-c. It is unclear how the authors made this conclusion as the signal (mislabeled tmp1?) does not appear to show zonal changes and there is no obvious zonation in the spatial UMAP. Stronger evidence and further explanation is required to how this conclusion was supported?

- Supplementary Figure 11: It is inappropriate to introduce more data in the discussion section of a manuscript.
- No reviewer token was provided for the shared datasets. It is hoped that these data are publicly accessible with rich metadata. The authors should also be depositing the bulk RNAseq data. It is not clear why this is not the case.

Reviewer #2 (Remarks to the Author):

The manuscript by Vandenberg and coworkers describe for the first time the impact of experimental breast cancer growth in mice on liver zonation using cutting-edge technologies. The work, although being merely observational, is outstanding in terms of novelty and the results are very intriguing, potentially providing the base for new investigations.

I here suggest some implementations that may further improve the quality of the manuscript and the 'usability' from the scientific community.

1 The current zonation partitioning is based on albumin and Cyp2e1. It would be nice to describe how the pattern is influenced by oxygen availability/hypoxia, according to the canonical liver zonation (see <https://pubmed.ncbi.nlm.nih.gov/28126520/>)

2 Consistently with the previous point, no information is given about the possible occurrence of anemia and consequent liver hypoxia in the tumor-bearing mouse model adopted in this study. Please add such information in the manuscript along with a brief discussion on the role of liver intrinsic vs extrinsic (e.g. anemia) factors on the disruption of liver zonation.

3 Is Saa expression coherent with Alb expression defining a 'secretory' area of the liver? Please check the availability of literature data and/or discuss.

4 The authors propose that proliferating neutrophil are found in the liver of tumor-bearing mice, not a common phenomenon. I suggest to clarify at least two points:

- what is the significance of such proliferation, given that the bone marrow looks functional, thus not requiring extramedullary granulopoiesis;

- are the markers used in the flow cytometry representative of neutrophils? The same have been used by others to identify myeloid-derived suppressor cells

(<https://pubmed.ncbi.nlm.nih.gov/30730772/>), a population of immature cells that frequently expands to support tumor immune escape.

5 The discussion section should better identify the significance of the observations reported in the complex systemic response of the host to tumor growth, maybe selecting few areas (e.g. systemic metabolism, inflammation, adaptive immune response) that are impacted by the alterations reported.

Point-by-point response to the reviewer's comments

We thank the reviewers for their constructive comments. All comments were to the point, and thus we were able to respond to them quickly. Below, please find our point-by-point responses to all comments and questions. We believe that we have examined each of the points raised and could respond thoroughly. As suggested, we have added new data and additional analysis. Again, we would like to thank all the reviewers for being interested in our manuscript and for their thoughtful suggestions, which have definitely made this a stronger paper.

List of the main changes:

1. We added data regarding the characteristics of the transplanted 4T1 tissues and their cancerous features *in vivo*
2. We added new data on the bone marrow of 4T1-bearing mice, showing a reduction in erythrocyte differentiation and heme biosynthesis, potentially leading to anemia
3. We made our single-cell and spatial transcriptomics data available in the DNA Data Bank of Japan, as well as on the Broad Institute Single Cell Portal.

Further replies to the reviewers' concerns are shown below. Our responses are marked in blue and red.

Reviewer #1 (Remarks to the Author):

This manuscript tested the hypothesis that distant tumors affect liver zonation. To accomplish this, breast cancer-bearing female mice were used following injection of 4T1 breast cancer cells. The authors identified a zonal hepatocyte trajectory by identifying spatial RNAseq data rich in Alb expression (denoting periportal hepatocytes) and Cyp2e1 expression (denoting pericentral hepatocytes). Hepatocytes were also resolved into treatment groups: sham-operated and cancer bearing. Results suggest gene expression for certain biological pathways, such triglyceride metabolism and aspartate biosynthesis were not zonally disrupted, while other pathways, such associated with xenobiotic metabolism, completely lose zonal distribution. Results from this study also highlight an influx in neutrophil infiltration in cancer bearing mice, with these infiltrating neutrophil populations showing gene expression signs of being immature. Taken together, the present study highlight cell-specific consequences in livers of 4T1 breast cancer bearing mice. The manuscript provides rich datasets that are thoroughly explored as well as presents novel hepatic cell-specific consequences due to breast cancer. The manuscript requires minor revisions to improve clarity and reproducibility.

Our response:

We appreciate the encouraging comments from this reviewer. “Hepatic cell-specific consequences due to cancer” precisely summarizes our current study. We would like to include such a sentence in our revised manuscript as follows. We thank this reviewer for precisely understanding our data and for insightful comments.

“Our data unravel cell type-specific consequences in the liver due to breast cancer transplantation.”
(Line 106-107 in the revised manuscript)

Please find our response to the comments by the reviewer below.

- The title would benefit from being more accurate to the work done such as using “Murine breast cancer cells disorganize ...” as the authors do not state whether solid tumors have formed. Breast cancer is often used instead of breast cancer in the manuscript. Is this truly a breast cancer model?

Our response:

The 4T1 breast cancer model is a commonly used syngeneic mouse cancer model (e.g., Pulaski and Ostrand-Rosenberg, *Curr. Protoc. Immunol.*, 2001). Transplanted tumor cells form cancerous tissues at the site of transplantation (e.g., mammary fat pad) and even colonize distant organs, forming metastasis. The invasive nature of this cancer cell line establishes this model as a cancer model. Histological data of this cancer model are published elsewhere (e.g., Candido et al., *Tumor Biol.*, 2014). In response to this comment, we include our own histology data of primary 4T1 cancer tissues (**Fig. R1a**). We confirmed the higher nuclei-to-cytoplasm ratio, anisonucleosis, and disruption in cellular polarity in the 4T1 tissues. These are typical signs of poorly differentiated cancer cells. In addition, our data detected a necrotic area most likely due to rapid proliferation. Infiltration of host immune cells was also observed. Furthermore, we provide the weight of 4T1 tissues recently recorded in our lab (**Fig. R1b**). The weight of 4T1 tissues increase over time (0.600 ± 0.11 g on day 7 after transplantation ($n=9$) and 1.40 ± 0.59 g on day 14 after transplantation ($n=18$)). 4T1 tissues are already visible by eye on day 7 after transplantation. On the basis of these, the murine 4T1 breast cancer model is an established transplantable cancer model. However, we admit that this is not a model where cancers originate from malignant mammary ductal cells, which might be the point of this comment. To clarify these points, we have added the following sentences in the revised manuscript.

“The 4T1 breast cancer model is a commonly used syngeneic mouse cancer model²⁶. Transplanted 4T1 cells are invasive, forming cancerous tissues in vivo.” (Line 115-117 in the revised manuscript)

Fig. R1 The basic characteristics of 4T1 cancer tissues

- (a) A frozen optimal cutting temperature compound (OCT) embedded section (thickness: 5µm) from 4T1 tissues obtained from 4T1-bearing mice on 14 days after transplantation. T: Tumor; N: Necrotic area; H: Hemorrhage
- (b) The weight of 4T1 cancer tissues on 7 and 14 days after transplantation. *n* = 9 for day 7 and *n* = 18 for day 14.

- The materials and methods section should include:
 - o Mouse strain details. Not provided when the model is first introduced and the sample size of treatment groups is not described. It is recommended that the authors review the ARRIVE guidelines (<https://arriveguidelines.org/>) and update the methods accordingly.

Our response:

We thank the reviewer for this comment. We have now summarized mouse strain details and the sample size information in the “Mice” section in the method as follows.

“All animal experiment protocols were approved by the Animal Care and Use committee of Kyoto University. Wild-type female BALB/c mice (8-week-old) purchased from Japan SLC Inc.

(Hamamatsu, Japan) were housed in a 12-hour light/dark paradigm with food ((CE-2, CLEA Japan, Inc., Tokyo) and water available *ad libitum*. No blinding was done in the experiments described in this study. Sample sizes were empirically determined. In scRNA-seq and spatial transcriptomics experiments, we acquired data from two mice in each experimental group. In other experiments, n = 4-5 was set as a threshold. These were reported in accordance with the *Animal Research: Reporting In Vivo Experiments* (ARRIVE) guidelines.” (Line 388-395 in the revised manuscript)

o It is unclear what the sham treatment is. Is it non-cancerous cells or just the vehicle or no manipulation?

Our response:

Thank you for pointing out this. The sham group receives RPMI1640 medium and the 4T1 group receives 4T1 cells in RPMI medium. We clarified this point in the method. The corresponding part is written in the “Cancer transplantation” section in the method.

“ 2.5×10^6 4T1 cells in 100 μ l of RPMI1640 medium or the same amount of cell-free RPMI1640 medium (sham) were inoculated subcutaneously into the right flank of 8-10-week-old BALB/c females and sacrificed 14 days after transplantation.” (Line 403-406 in the revised manuscript)

o What was the confluency of the cultured cells prior to injection?

Our response:

The confluency of 4T1 cells was approximately 70-80% prior to injection. We clarified this point in the method. The corresponding part is written in the “Cancer transplantation” section in the method.

“The confluency of 4T1 cells was approximately 70-80% prior to injection.” (Line 406-407 in the revised manuscript)

o What were the revision versions for the 10X Genomics user guides cited throughout the methods. For reproducibility 10X Genomics numbers their revisions typically as CG[number].

Our response:

We agree that clarifying these details in the method is important for readers. According to this comment, we have added the versions for the 10x Genomics user guides as follows.

“Libraries for Visium were prepared according to the Visium Spatial Gene Expression User Guide (catalog no. CG000239 Rev A, 10xGenomics).” (Line 414-416 in the revised manuscript)

“The optimal permeabilization time for 10 μ m thick liver sections was determined to be 6 minutes using 10x Genomics Visium Tissue Optimization Kit (catalog no. CG000238 Rev A, 10x Genomics).” (Line 420-422 in the revised manuscript)

“The single-cell RNA-seq library was constructed by using the Chromium Controller and Chromium Single Cell 3' Reagent Kits v3.1 (10x Genomics) following the standard manufacturer's protocols (catalog no. CG000227 Rev C).” (Line 494-496 in the revised manuscript)

o Can the authors provide the list of filtered GO terms as supplemental material or using a data repository (e.g., Dataverse) for reproducibility? This will also assist readers in knowing that genes were considered in calculating pathway scores.

Our response:

We appreciate this comment. We included supplementary Table S2 and S3 that list the filtered GO terms. We also added a new Table S1 that summarizes the set of genes included in each GO term.

o How were the pathway scores calculated? It is only stated that they were calculated and differences were determined by WRS test.

Our response:

In the Methods section we mentioned that we used the Seurat package AddModuleScore for this calculation. We have added a few lines of explanation, although we refer to the original paper for more details. This part is written in the “Analysis of zonation of genes sets and their module scores” section.

“In brief, modules scores reflect the average expression of the set of genes within each spot or cell, subtracted by the average expression of a number of control genes with similar average expression levels as the input set. AddModuleScore was used using default parameters; all genes were divided into 24 bins based on their expression level ($n_{bin} = 24$) and for each input gene 100 genes were randomly selected from the same bin ($ctrl = 100$) to serve as a control set.” (Line 459-464 in the revised manuscript)

o There is an overall lack of information on the bulk RNAseq. How was RNA isolated? was the RNA QC'd for RIN value? how were libraries prepared? what were the sequencing parameters? specify the reference genome?

Our response:

We apologize for this confusion. We want to clarify that the bulk RNA-seq data comes from our previous datasets (i.e., we re-analyzed our datasets). This is why our manuscript did not contain the methodology information for this data. We should have better clarified this point in the original manuscript. Our previous sentence regarding this issue was unclear.

Just for your information, total RNAs were extracted using the RNeasy mini kit, and RIN values determined by the 2100 Bioanalyzer system were as follows (**Table R1**). The obtained reads were mapped to the mouse genome version mm10. The raw sequencing data is available in the DNA Data Bank of Japan (DDBJ) under the accession number DRA01192.

sample name	sample title	Run	Biosample Accession	RNA Concentration [ng/ μ l]	rRNA Ratio [28s / 18s]	RNA Integrity Number (RIN)
L1	WT_Sham_1	DRR289380	SAMD00317733	271	1.4	9.5
L2	WT_Sham_2	DRR289381	SAMD00317734	162	1.4	9.6
L3	WT_Sham_3	DRR289382	SAMD00317735	283	1.4	9.6
L4	WT_Sham_4	DRR289383	SAMD00317736	102	1.2	9.4
L5	WT_4T1_1	DRR289384	SAMD00317737	412	1.5	9.6
L6	WT_4T1_2	DRR289385	SAMD00317738	566	1.5	9.5
L7	WT_4T1_3	DRR289386	SAMD00317739	464	1.5	9.6
L8	WT_4T1_4	DRR289387	SAMD00317740	415	1.5	9.5

Table R1. The QC data for the bulk liver transcriptome samples

To clarify these points, we added the following sentences in the revised manuscript.

“This trend was validated using bulk RNA-seq datasets that we previously reported⁸, suggesting that breast cancer transplantation repressed metabolism in the liver (**Fig. S6**).” (Line 198-199 in the revised manuscript)

“The bulk transcriptome data from the liver and bone marrow are also available (DRA01192⁸ and DRA015216, respectively).” (Line 607-608 in the revised manuscript)

o No QC metric provided for qPCR RNA input.

Our response:

We have added the quality information of qPCR RNA input (**Table S5**) in the revised manuscript.

“Quality check data for our qPCR are provided in Table S5.” (Line 584 in the revised manuscript)

- The authors do not describe whether administration of cancer cells produced the anticipated phenotype. Did tumors develop and grow as expected in this model?

Our response:

As mentioned earlier, administration of cancer cells produced the anticipated phenotype.

“The formation of primary cancer tissues was confirmed by eye upon sacrificing mice.” (Line 407-408 in the revised manuscript)

- The following figures need to revision for clarity and interpretation.
 - o In the figures where the Visium slides are shown the size is so small that they look like dark squares. It may be more useful to move some of the figures as supplemental for easier visualization. The authors should also strongly consider uploading both the spatial and single-cell/nuclei data to the Broad Single Cell Portal for a more interactive visualization and provide the link in this publication. Several examples are available on how to deposit both types of data.

Our response:

Thank you for this constructive comment. We have accordingly made the colors easier to see in the figures showing Visium slices, by increasing the size of spots and decreased the width of the black circle around each spot. We also increased the size of the Visium figures, where possible. In addition, in response to this comment, we uploaded our data to the Broad Institute Single Cell Portal under the accession numbers of **SCP2045 (scRNA-seq)** and **SCP2046 (Visium)**. It was good for us to know that such a convenient database exists. We again thank this reviewer for this important comment.

- o Several figures are missing the label for interpretation or use a generic “value” term which make it very difficult to interpret. Please use appropriate labels such as “activity score” instead of “value” and denoting what the units represent for intensity in the Visium figures.

Our response:

We appreciate this comment. In the revised manuscript, we indicated more clearly what color gradients are showing, within the figures themselves and/or in the figure legends.

o Most of the Visium plots appear to be mislabeled as showing the expression of *tmp1* which in addition to not being correct, doesn't follow normal mouse nomenclature (should be capitalized and italicized).

Our response:

We thank this reviewer for pointing out this. We have removed the erroneous “*tmp1*” labels, and we have corrected the gene nomenclature.

o Have the authors examined alternative means to plot spatial expression as a continuum across bins to improve representation of spatially resolved expression?

Our response:

Yes, we have considered several options. In the manuscript we used three ways to show the spatially resolved expression:

- The most intuitive way is to show the Visium spots on top of the tissue slices. This is good for patterns that are clearly spatially resolved, such as *Alb* and *Cyp2e1*.
- We also showed the same spots in a UMAP plot where there was a clear association between UMAP 1 and the main spatial feature of interest (periportal to pericentral location in the tissue). In this visualization, even weaker spatial patterns are easy to spot.
- When we want to show the spatial pattern of many genes (or sets of genes) we used the binned heatmaps, where bins show the distance to the *Alb*^{high} or *Cyp2e1*^{high} spots. Although this is less direct, it does allow us to give an overview of patterns easily.

Although approaches 1 & 2 are pretty intuitive, they are not suitable for showing the pattern of many genes or gene sets. A fourth alternative is to show the signal not using colors (as in the heatmap) but in a line plot (as an example we refer to the figure in our response to the next comment below (**Fig. R2**)).

We have also considered scatter plots showing the correlation of expression of genes with *Alb* in sham samples (X axis) and cancer-bearing samples (Y axis). Although such a plot also can give a general overview, we found that it was not intuitive at all, and we decided not to use it in this manuscript.

- The authors conclude identification of zonally resolved activated macrophage citing supplementary Figure 10a-c. It is unclear how the authors made this conclusion as the signal (mislabeled *tmp1*?) does not appear to show zonal changes and there is no obvious zonation in

the spatial UMAP. Stronger evidence and further explanation is required to how this conclusion was supported?

Our response:

We appreciate this comment. Indeed, the macrophage zonation appears less clear compared to e.g., neutrophils. Yet, “positive regulation of macrophage activation” is high in *Alb*^{high} spots compared to *Cyp2e1*^{high} spots. We made this clearer by putting a circle in the UMAP plot (**Fig. S11 in the revised manuscript**), indicating spots of the cancer-bearing samples with low module scores. Here, in our response, we also provide a line plot to indicate the degree of zonation of this pathway (**Fig. R2**; red arrow roughly corresponds to the spot indicated in the UMAP plot (**Fig. S11b**)). We chose not to put this plot in the revised manuscript because we want to avoid showing the same data in too many alternative ways. We hope indicating the spots using a circle in the UMAP plot will be sufficient to draw attention to the zonation pattern.

Fig. R2 Zonated macrophage gene expression
Mean module scores of genes associated with “positive regulation of macrophage activation” in function of distance to *Alb*^{high} (left side) and *Cyp2e1*^{high} (right side) in spots of the sham (black) and cancer-bearing (grey) Visium samples. The red arrow roughly indicates the same spots as indicated in red in **Fig. S11b**.

- Supplementary Figure 11: It is inappropriate to introduce more data in the discussion section of a manuscript.

Our response:

We appreciate this comment. We now introduce **Fig. S12** (originally Fig. S11) as follows in the “Results” section.

“Our data revealed other zonated patterns of gene expression in the livers of cancer-bearing mice whose significance is relatively unclear to us at present (**Fig. S12**). The transcytosis pathway was much more active in *Alb*^{high} zones rather than in *Cyp2e1*^{high} zones (**Fig. 4a** and **Fig. S12a-c**). 4T1 cancer transplantation elevated the expression of genes involved in this pathway, potentially enhancing transcytosis in endothelial cells nearby portal veins. In an interesting contrast, the

transepithelial transport appeared zonated towards *Cyp2e1*^{high} zones and activated in the presence of 4T1 breast cancers (**Fig. 4a** and **Fig. S12d-f**). A previous study reported immune cell zonation orchestrated by liver sinusoidal endothelial cells²³. In this report, the authors suggested that liver sinusoidal endothelial cells sense the microbiome, accordingly modulating immune cell localization. The complex interaction among different cell types and stimuli might underlie the altered zonation of immune cells and endothelial cells in our experimental settings. Addressing these zonation patterns in detail is also an important next step.” (Line 281-293 in the revised manuscript)

- No reviewer token was provided for the shared datasets. It is hoped that these data are publicly accessible with rich metadata. The authors should also be depositing the bulk RNAseq data. It is not clear why this is not the case.

Our response:

We thank this reviewer for this important comment. As written in the “Data availability statement” and this point-by-point response, the following data are publicly available in the DNA Databank of Japan (DDBJ).

Visium: DRA014802

scRNA-seq: DRA009332

Bulk liver RNA-seq: DRA01192 (this data had been published in Mizuno et al., 2022)

Bone marrow RNA-seq: DRA015216 (newly added in this revision)

As mentioned earlier, the scRNA-seq and Visium data are also available in the Broad Institute Single Cell Portal under the accession numbers of SCP2045 (scRNA-seq) and SCP2046 (Visium).

Reviewer #2 (Remarks to the Author):

The manuscript by Vandebon and coworkers describe for the first time the impact of experimental breast cancer growth in mice on liver zonation using cutting-edge technologies. The work, although being merely observational, is outstanding in terms of novelty and the results are very intriguing, potentially providing the base for new investigations.

I here suggest some implementations that may further improve the quality of the manuscript and the ‘usability’ from the scientific community.

Our response:

We appreciate the encouraging and valuable comments from this reviewer. We hope that this manuscript will be a basis for a new direction of investigations of cancer’s adverse effects on the host.

1 The current zonation partitioning is based on albumin and Cyp2e1. It would be nice to describe how the pattern is influenced by oxygen availability/hypoxia, according to the canonical liver zonation (see <https://pubmed.ncbi.nlm.nih.gov/28126520/>)

Our response:

We thank this reviewer for bringing up an important subject. To this end, we investigated the correlation between oxygen availability and the expression patterns of biological pathways according to a reference (Kietzmann, *Redox Biol.*, 2017: **Fig. S3** in the revised manuscript). We found that a set of pathways correlated with oxygen availability showed such zonation in our datasets including “Gluconeogenesis,” “Urea metabolic process,” and “Glutamine family amino acid biosynthetic process.” These pathways appeared active where oxygen is rich (i.e., *Alb^{high}* spots). We also found pathways being active where oxygen is poor (i.e., *Cyp2e1^{high}* spots). These pathways included the “Xenobiotic catabolic process” and “Bile acid biosynthetic process.” These results indicate that our datasets captured oxygen-associated zoned gene expression patterns in the liver. We summarized these data in **Fig. S3** in the revised manuscript and introduced them as follows.

“We found that a set of pathways that have been reported to be correlated with oxygen availability²⁸ showed such zonation in our datasets, including “gluconeogenesis”, “urea metabolic process”, and “glutamine family amino acid biosynthetic process” (**Fig. S3**). These pathways appeared active where oxygen is rich (i.e., *Alb^{high}* spots). We also found pathways being active where oxygen is poor (i.e., *Cyp2e1^{high}* spots). These pathways included the “xenobiotic catabolic process” and “bile acid biosynthetic process.” (Line 140-145 in the revised manuscript)

We also examined hypoxia-related genes such as *Hif1 α* . However, the expression levels of these genes were very low in our Visium datasets, and we could not find a zoned expression pattern.

2 Consistently with the previous point, no information is given about the possible occurrence of anemia and consequent liver hypoxia in the tumor-bearing mouse model adopted in this study. Please add such information in the manuscript along with a brief discussion on the role of liver intrinsic vs extrinsic (e.g. anemia) factors on the disruption of liver zonation.

Our response:

Thanks to this comment, we became aware of recent progress in understanding cancer-associated anemia (e.g., Furrer et al., *Sci. Adv.*, 2022). This insightful comment led us to investigate our unpublished RNA-seq data sets from the bone marrow of sham and 4T1-bearing mice (**Fig. S13**). We found that 4T1 transplantation induces various gene expression changes in the bone marrow (**Fig. S13a**). Gene ontology analyses and GSEA suggested the downregulation of erythrocyte differentiation and heme metabolism (**Fig. S13b, c**). Bone marrow looked red in the sham group. Most importantly, such red color became relatively faint in the 4T1 group, suggesting that erythrocytes decreased upon 4T1 transplantation (**Fig. S13d, e**). These results were consistent with a recent related report (Furrer et al., *Sci. Adv.*, 2022). On the basis of these, we concluded that 4T1-bearing mice likely suffer from anemia, which might be an important extrinsic factor in the disruption of liver zonation. We next examined if 4T1 transplantation induces hypoxia in the liver. GSEA using the bulk RNA-seq datasets demonstrated that 4T1 transplantation did not induce hypoxia in the liver (**Fig. R3a**). We then looked into the expression of typical hypoxia markers (Puente-Santamaria et al., *BMC Bioinformatics*, 2022). We found that the expression of *Car9*, *Vegf α* , *Hif1 α* , and *Kdr* was not strongly affected in the cancer-bearing samples (**Fig. R3b**). It is therefore unlikely that 4T1 cancer transplantation causes strong hypoxia in the liver. However, these data do not exclude the possibility that the livers of 4T1-bearing animals suffer from a milder level of hypoxia. Given the complexity of this topic, we chose not to include these hypoxia data in the manuscript. We have added the following sentences in the revised manuscript. We thank this reviewer for this valuable comment.

“It is also important to clarify the involvement of liver extrinsic factors in the disruption of liver zonation. It is known that cancers cause anemia³², which might be an extrinsic factor. To test if cancers cause anemia phenotypes in our experimental settings, we analyzed the bone marrow of sham and 4T1-bearing mice. We found that 4T1 transplantation induces various gene expression changes in the bone marrow (**Fig. S13a**). Gene ontology analyses and GSEA suggested the downregulation of erythrocyte differentiation and heme metabolism (**Fig. S13b, c**). Bone marrow

looked red in the sham group. Such red color became relatively faint in the 4T1 group, suggesting that erythrocytes decreased upon 4T1 transplantation (Fig. S13d, e). On the basis of these, we concluded that 4T1-bearing mice likely suffer from anemia, which might be an important extrinsic factor in the disruption of liver zonation.” (Line 294-303 in the revised manuscript)

Fig. R3 The effects of 4T1 transplantation on liver hypoxia

- (a) Gene set enrichment analysis (GSEA) against differentially expressed genes in the livers of 4T1-bearing animals ($n = 4$; bulk RNA-seq). Enrichment score (ES), normalized enrichment score (NES), and adjusted p value (Padj) are shown.
- (b) The expression of typical hypoxia-related genes in the livers of sham and 4T1-bearing mice (bulk RNA-seq). Data are normalized to the average of sham and presented as mean \pm s.e.m.

3 Is *Saa* expression coherent with *Alb* expression defining a ‘secretory’ area of the liver? Please check the availability of literature data and/or discuss.

Our response:

As shown in Fig. S7d, e, *Saa* expression appeared coherent with *Alb* expression. Thanks to this reviewer’s comment, we realized that *Saa* expression is biased toward “secreting” hepatocytes. We found publications that mention zoned protein secretion in the portal layers (e.g., Manco and Itzkovitz, *J. Hepatology*, 2021; Paris and Henderson, *Hepatology*, 2022). Accordingly, we added the following sentences in the revised manuscript.

“It is likely that *Saa* expression is biased toward secreting hepatocytes, which is consistent with zoned protein secretion in the portal layers^{14,30}.” (Line 324-326 in the revised manuscript)

4 The authors propose that proliferating neutrophil are found in the liver of tumor-bearing mice, not a common phenomenon. I suggest to clarify at least two points:

- what is the significance of such proliferation, given that the bone marrow looks functional, thus not requiring extramedullary granulopoiesis;

Our response:

Thank you for this important comment. At this point, whether neutrophils indeed proliferated in the liver remains unclear. It is possible that neutrophils proliferated in the bone marrow and immediately migrated to the liver, with relatively higher, residual *Mki67* expression. Moreover, as shown in **Fig. S13** in our revised manuscript, the bone marrow of 4T1-bearing animals exhibited a series of anomalies. We summarized these points in the revised manuscript as follows.

“It remained unclear if neutrophils proliferated in the liver. We assumed that *Mki67* expression in neutrophils in the liver might be remnants of neutrophil proliferation in the bone marrow.” (Line 242-244 in the revised manuscript)

- are the markers used in the flow cytometry representative of neutrophils? The same have been used by others to identify myeloid-derived suppressor cells

(<https://pubmed.ncbi.nlm.nih.gov/30730772/>), a population of immature cells that frequently expands to support tumor immune escape.

Our response:

We appreciate this comment. The markers we used in this study are representative of neutrophils. Indeed, as described in this review article (Veglia et al., Nat. Rev. Immunol., 2021) and pointed out by this reviewer, myeloid-derived suppressor cells are considered pathologically activated neutrophils and monocytes with potent immunosuppressive activity. Given that our study does not investigate the immunosuppressive roles of detected neutrophils, we want to call them simply (immature) neutrophils. At the same time, we added our discussion regarding myeloid-derived suppressor cells in the revised manuscript as follows.

“These immature neutrophils might represent myeloid-derived suppressor cells (i.e., pathologically activated neutrophils and monocytes)²⁹.” (Line 254-255 in the revised manuscript)

5 The discussion section should better identify the significance of the observations reported in the complex systemic response of the host to tumor growth, maybe selecting few areas (e.g. systemic metabolism, inflammation, adaptive immune response) that are impacted by the alterations reported.

Our response:

We thank this reviewer for this suggestion. As pointed out by this reviewer, cancers cause various systemic effects on the host. These alterations likely interact with each other. For example, anemia could be an extrinsic factor for the disruption of liver zonation. On the other hand, the massive downregulation of e.g., triglyceride metabolism should affect non-hepatocyte cell types within and outside the liver, potentially further altering the immune system status. Such response of the host to cancer growth in turn may benefit cancers. According to these, we have added the following sentences in the revised manuscript.

“Potential interaction among cancer-induced abnormalities

Cancers cause various systemic effects on the host¹⁻¹¹. These alterations likely interact with each other. For example, anemia could be an extrinsic factor for the disruption of liver zonation (**Fig. S13**). On the other hand, the massive downregulation of e.g., triglyceride metabolism should affect non-hepatocyte cell types within and outside the liver, potentially altering the immune system status. These abnormalities of the host might benefit cancer growth. Dissecting such complex interactions at the body-wide level is critical to understand cancer’s adverse effects on the host.” (Line 372-379 in the revised manuscript)

REVIEWERS' COMMENTS:

Reviewer #1 (Remarks to the Author):

The authors have comprehensively responded to our concerns and revised the manuscript accordingly. I have no further comments or concerns.

Reviewer #2 (Remarks to the Author):

It is a pleasure to re-revise this manuscript. All of my suggestions have been taken into account both providing a satisfactory response and improving the manuscript accordingly. I noticed that the same appened with the second reviewer that focused more on technical aspects of the research, thus complementing the points raised by me.